# Mineral-Doped Poly(L-lactide) Acid Scaffolds Enriched with Exosomes Improve Osteogenic Commitment of Human Adipose-Derived Mesenchymal Stem Cells

**DOI:** 10.3390/nano10030432

**Published:** 2020-02-29

**Authors:** Maria Giovanna Gandolfi, Chiara Gardin, Fausto Zamparini, Letizia Ferroni, Micaela Degli Esposti, Greta Parchi, Batur Ercan, Lucia Manzoli, Fabio Fava, Paola Fabbri, Carlo Prati, Barbara Zavan

**Affiliations:** 1Laboratory of Biomaterials and Oral Pathology, School of Dentistry, Department of Biomedical and Neuromotor Sciences, University of Bologna, 40125 Bologna, Italy; fausto.zamparini2@unibo.it (F.Z.);; 2Medical Sciences Department, University of Ferrara, 44100 Ferrara, Italy; 3Department of Civil, Chemical, Environmental and Materials Engineering, University of Bologna, 40136 Bologna, Italy; 4Department of Metallurgical and Materials Engineering, Middle East Technical University (METU), 06800 Ankara, Turkey; 5Biomedical Engineering Program, METU, 06800 Ankara, Turkey; 6BIOMATEN, Center of Excellence in Biomaterials and Tissue Engineering, METU, 06800 Ankara, Turkey; 7Cellular Signaling Laboratory, Institute of Human Anatomy, Department of Biomedical and Neuromotor Sciences, University of Bologna, 40126 Bologna, Italy; 8Endodontic Clinical Section, School of Dentistry, Department of Biomedical and Neuromotor Sciences, University of Bologna, 40125 Bologna, Italy

**Keywords:** exosomes, human adipose mesenchymal stem cells (hAD-MSCs), polylactic acid scaffold, calcium silicates, exosome enriched scaffolds, regenerative bone healing, green bioplastics, bioresorbable scaffold, natural nanovesicles, nanodelivery, personalized regenerative medicine

## Abstract

Exosomes derived from mesenchymal stem cells are extracellular vesicles released to facilitate cell communication and function. Recently, polylactic acid (PLA), calcium silicates (CaSi), and dicalcium phosphate dihydrate (DCPD) have been used to produce bioresorbable functional mineral-doped porous scaffolds-through thermally induced phase separation technique, as materials for bone regeneration. The aim of this study was to investigate the effect of mineral-doped PLA-based porous scaffolds enriched with exosome vesicles (EVs) on osteogenic commitment of human adipose mesenchymal stem cells (hAD-MSCs). Two different mineral-doped scaffolds were produced: PLA-10CaSi-10DCPD and PLA-5CaSi-5DCPD. Scaffolds surface micromorphology was investigated by ESEM-EDX before and after 28 days immersion in simulated body fluid (HBSS). Exosomes were deposited on the surface of the scaffolds and the effect of exosome-enriched scaffolds on osteogenic commitment of hAD-MSCs cultured in proximity of the scaffolds has been evaluated by real time PCR. In addition, the biocompatibility was evaluated by direct-contact seeding hAD-MSCs on scaffolds surface-using MTT viability test. In both formulations, ESEM showed pores similar in shape (circular and elliptic) and size (from 10–30 µm diameter). The porosity of the scaffolds decreased after 28 days immersion in simulated body fluid. Mineral-doped scaffolds showed a dynamic surface and created a suitable bone-forming microenvironment. The presence of the mineral fillers increased the osteogenic commitment of hAD-MSCs. Exosomes were easily entrapped on the surface of the scaffolds and their presence improved gene expression of major markers of osteogenesis such as collagen type I, osteopontin, osteonectin, osteocalcin. The experimental scaffolds enriched with exosomes, in particular PLA-10CaSi-10DCPD, increased the osteogenic commitment of MSCs. In conclusion, the enrichment of bioresorbable functional scaffolds with exosomes is confirmed as a potential strategy to improve bone regeneration procedures.

## 1. Introduction

Bone defects and mainly oral bone defects are widely present in the world population, for a several number of pathological conditions, such as periapical bone lesions, cysts, tumor, bone atrophies, and bone trauma [1,2,3]. Bone tissue, unlike other tissues possesses the possibility to heal and remodel [4,5,6,7].

Exosomes-phospholipid bilayer nanovesicles (50–100 nm) containing lipids, nucleic acids, proteins and signaling molecules [8] are released from all cell types to facilitate the cells communication and are able to modify the activities of target cells by means of proteins, growth factors, mRNAs, and micro RNAs internalization [9]. Exosomes are natural mediators in cell–cell communication involved in several physiological processes including neo-angiogenesis, regulation of immune response and extracellular matrix remodeling, affecting cell phenotype, recruitment, proliferation and differentiation [10]. These vesicles have a great role in critical bone defects healing, impacting on tissue microenvironment and promoting neo-angiogenesis events [11] and the homeostasis of osteoblast-osteoclasts cells [12].

The use of natural nanovesicles as a shuttle of therapeutic molecules with pro-regenerative properties may reveal a significant interest in personalized regenerative medicine, as these vesicles may be isolated from the same patient, avoiding the problem of immunogenicity, which is one of the most important complication of cell-based therapy [10,13]. There is evidence demonstrating that heterologous exosomes released by mesenchymal stem cells may be considered as a reliable and safe source for therapeutic exosomes. However, the use of an autologous approach cannot be excluded [13]. In this context, mesenchymal stem cells and monocyte derived stem cells represent the most common safe and valuable source for therapeutic exosomes [13].

Several attempts have been performed to design an ideal biomaterial to surpass the traditional autologous bone grafts procedures, which are highly expensive, invasive, with risks of complications and with variable graft resorption rate [13,14].

One of the most promising experimental strategy to achieve oral bone regeneration is through a biocompatible 3D porous scaffold [15]. A scaffold should induce a positive interaction with the surrounding bone tissues, by acting as a support for mineralizing cells, ensuring an adequate environment for growth factors and nutrients, and being able to stimulate a proper communication among the resident mineralizing cells [16].

In this context, scaffold surface porosity [17] and the related micro and nano-morphology [17,18] directly influence the cell’s behavior. Bone regeneration procedures need a constant flow of nutrients, cells, and growth factors from the outer portion to the core of the scaffold [11,19]. Currently, there is no consensus concerning the more appropriate porosity value or pore size. However, when mechanical properties are satisfied, high porosity values (over 90%) are recommended [17], with a wide range of pore size from 10 to at least 200 µm [20].

Several scaffolds have been proposed in literature, however none of these demonstrated all of these properties [21,22,23]. Therefore, multiple materials have been combined in order to limit the single material drawbacks and to gain adequate characteristics [21,22,23]. Biodegradable synthetic polymers show the advantage of the easy modification of their composition with following change of their chemical, physical and mechanical properties [24,25]. In biomedical field, these materials are currently used alone or in combination with devices for bone fracture repair, ligament reconstruction, or surgical dressings [26,27].

Polylactic acid (PLA) is a green bioplastic synthetically prepared through the polymerization of bacteria-derived lactic acid. Chemically, PLA is a thermoplastic semicrystalline aliphatic polyester with a linear polymeric chain, composed of a mixture of the two lactide isomers of lactic acid which are then polymerized into PLA [21,22,28]. PLA is naturally produced from bacterial fermentation of polysaccharide-based renewable natural resources [29,30,31] and shows numerous advantages compared to other synthetic polymers, including significant energy savings, recyclability, and ability to tailor physical properties [32]. PLA is widely used to produce a number of biomaterials for several surgical branches (as sutures, stents, screws, or other fracture fixation devices) for its properties of biocompatibility and non-toxicity [33]. In oral surgery, PLA and its copolymers have been marketed as membranes or as temporary space fillers to be placed after ridge preservation procedures, in presence of post-extractive sockets, or after periodontal surgeries [34].

Several attempts have been made in order to design polylactic-based scaffolds blended with other co-polymers or bioactive molecules for bone tissue engineering, using different fabrication techniques [35,36,37]. Recently, experimental PLA scaffolds [36] or PLA-based scaffolds containing chitosan [35] or calcium deficient hydroxyapatite [37] produced by 3D printing [36,37] or electrospinning technique [35] have been proposed for experimental bone regeneration procedures.

Calcium silicate (CaSi) are biointeractive biocompatible materials widely used in dentistry where the regeneration of mineralized tissues (bone, dentine) is required [38,39]. Their triggering activity on mineralizing cells — as osteoblasts [40], cementoblasts [41], pulp cells [42], orofacial bone mesenchymal stem cells [43], marrow stromal cells [44], mesenchymal stem cells derived from human periapical cyst [45]—is correlated to their ability to release biologically active ions (OH^-^ and Ca^++^ ions) [46,47,48] and to nucleate apatite [44,49,50]. The formation of apatite—by the adsorption of Ca and P ions on the silanol groups (Si-OH) and the precipitation of calcium phosphates which mature into a B-type carbonated-apatite [44,49,50]—provides an excellent stimulus for bone regeneration and bone bonding. CaSi are able to stimulate new bone formation when positioned in periapical bone defects [38,39,48,51].

Therefore, the addition of bioactive CaSi on a PLA matrix could represent an attractive strategy to improve the biological properties and favor gene commitment for regenerative purposes. Actually, PLA scaffolds doped with bioactive CaSi recently showed the absence of toxicity toward fibroblasts [52] and mesenchymal stem cells from human periapical cyst [39], high values of bulk porosity, adequate thermal-mechanical properties, the ability to release Ca^++^ and OH^−^ and nucleate apatite [52], resulting in promising materials in regenerative dentistry.

For these reasons, the aim of the present work is to evaluate the association of experimental PLA-based scaffolds containing different amounts of CaSi and DCPD with exosomes as innovative tool direct to improve bone regeneration acting on osteogenic properties of human adipose mesenchymal stem cells.

## 2. Materials and Methods

### 2.1. Scaffolds Preparation

Natureworks poly(L-lactic acid) (PLA; IngeoTM biopolymer PLA 4060D, Natureworks LLC, Blair, NE, USA) MW = 65,000 g/mol.

Methanol (MeOH), ethanol (EtOH, 99.8%), 1,4-dioxane (DIOX) and chloroform (CHCl_3_, HPLC grade) from Sigma Aldrich (Milan, Italy) were used.

PLA was purified via dissolution in CHCl_3_ (10% wt/vol) and then reprecipitated in pre-cooled MeOH in order to completely remove any traces of polymerization catalysts.

Dicalcium phosphate dihydrate (DCPD; CaHPO_4_·2H_2_O) powder (Sigma-Aldrich, Steinheim, Germany) and/or calcium silicate (CaSi) powder (Aalborg, Denmark), obtained by melt-quenching technique and milling procedures, and composed of dicalcium silicate, tricalcium silicate, tricalcium aluminate and calcium sulfate were added to PLA [52,53]. The scaffolds were prepared by thermally induced phase-separation technique previously described [52,53].

Two different disks formulations (diameter 60 ± 1 mm, thickness 10 ± 0.1 mm), namely PLA-5CaSi-5DCPD and PLA-10CaSi-10DCPD containing 5% or 10% by weight of mineral powders, were obtained.

The scaffolds were designed and produced at the University of Bologna.

### 2.2. ESEM-EDX Analysis for Morphometric Analysis and Surface Porosity after 28 Days Soaking in Simulated Body Fluid

Mineral-doped scaffolds were cut into samples (10.00 ± 1.00 mm height, 15.00 ± 1.00 mm length, and 10.00 ± 1.00 mm width; n = 3 per formulation) and aged in 20 mL of Hank’s balanced saline solution (HBSS) at 37 °C for 28 days [48,50]. The HBSS composition was: Mg^++^ 0.81 mM, K^+^ 5.8 mM, Ca^++^ 1.27 mM, Na^+^ 141.6 mM, Cl^−^ 144.7 mM, HCO_3_^−^ 4.17 mM, SO_4_^−^ 0.81 mM, H_2_PO_4_^−^ 0.44 mM, and HPO_4_^−^ 0.336 mM. Surface modification was investigated using HBSS as simulated body fluids—following ISO 23317 (implants for surgery—in vitro evaluation for apatite-forming ability of implant materials) method, which evaluates layers precipitated on the materials soaked in simulated body fluid—in agreement with previous studies evaluating the apatite forming ability and the bioactivity of biomaterials [46,48,50,52,54]. HBSS has been selected in order to have a commercially available standardized soaking medium mimicking the composition of inorganic ions of human blood plasma (HBSS contains a lower calcium concentration and same phosphate concentration in comparison with human plasma).

The surface of the scaffolds was examined, before (fresh samples) and after 28 days aging in HBSS, using an environmental scanning electron microscope (ESEM; Zeiss EVO 50, CarlZeiss, Oberkochen, Germany) connected to a secondary electron detector for energy dispersive X-ray analysis (EDX; Oxford INCA 350 EDS, Abingdon, UK) using computer-controlled software (Inca Energy Version 18). Uncoated samples were examined using a quadrant back-scattering detector at 100 Pa low-vacuum, working distance 8.5 mm, 133 eV resolution, 0.5% wt detection level, 100 µs amplification time, 60 s measuring time.

ESEM images were analyzed using Image J program (National Institutes of Health, Bethesda, USA) to evaluate the surface porosity of the scaffolds calculated as the ratio between the blackest areas (micropores) and the total examined area [51,55].

For each scaffold formulation the measurements were repeated three times in the same area and the mean values were recorded. The evaluations were performed at 100×, 500×, and 1000× magnifications.

Surface porosity variations among the fresh scaffolds and the 28 d aged scaffolds was analyzed using two-way ANOVA followed by RM Student–Newman–Keuls test (*p* value was set at 0.05).

### 2.3. Cell Culture

Human adipose mesenchymal stem cells (hAD-MSCs) (Lonza Inc., Walkersville, MD, USA) were plated in culture flasks (Sigma-Aldrich, Milan, Italy) at a density of 10  ×  10^3^ cells/cm^2^ with 15  mL of dedicated growth medium (MSCGM™; Bullet Kit, Lonza Inc., Walkersville, MD, USA) and incubated at 37 °C in humid environment supplemented with 5% CO_2_. hAD-MSCs were then passaged at 80% confluence in a ratio 1:3 in trypsin/EDTA solution (Gibco, Life Technologies, Carlsbad, CA, USA), the medium refreshed two times a week.

### 2.4. hAD-MSCs Viability by Tetrazolium Salt (MTT) Assay

In order to test the biocompatibility of the experimental scaffolds, cells have been seeded onto the scaffold surface and cultured up to 10 days in dedicated growth medium MSCGM. The test of viability has been performed following ISO 10993 Biological evaluation of medical devices. For each sample, the culture medium was removed and cells were incubated at 37 °C for 3 h in 1 mL of MTT solution (0.5 mg/mL MTT in PBS). Then, MTT solution was removed and the remaining MTT solution was extracted with 0.5 mL of 10% dimethyl sulfoxide in isopropanol for 30 min at 37 °C. For each sample the optical density (OD) values were recorded in duplicate at 570 nm, in 200-μL aliquots using a multilabel plate reader (Victor 3, Perkin Elmer, Milan, Italy).

Rectangular bottom plastic flasks, manufactured from optically clear polystyrene, sterilized by gamma irradiation and certified nonpyrogenic were used as control.

### 2.5. Exosomes Isolation from Stem Cells

EVs isolation was performed from the growth medium MSCGM™ of hAD-MSCs. Then, hAD-MSCs (10 × 10^6^ cells, passages 4–6) were cultured in 75 cm^2^ tissue flasks to reach 40% confluence; then the culture medium was changed and the cells were incubated for additional 48–72 h to reach the confluence of 70–80%. Cell culture supernatants were collected and centrifuged at 4 °C for 10 min at 200 g and then for 10 min at 500 g. Culture supernatants were thawed and spun down vertically at 4 °C for 20 min at 2000*g* and then centrifuged horizontally at 100,000*g* for 75 min. Finally, the supernatant was discarded and the exosome pellet was re-suspended in 1 mL of PBS.

### 2.6. Exosome Labeling with Red Fluorescent

EVs isolation was performed from the growth medium MSCGM™ of hAD-MSCs. In detail, hAD-MSCs (10 × 10^6^ cells, passages 4–6) were cultured in 75 cm^2^ tissue flasks to reach 40% confluence; the culture medium was changed and the cells were incubated for additional 48–72 h to reach the confluence of 70–80%. Cell culture supernatants were collected and centrifuged at 4 °C for 10 min at 200 g and then for 10 min at 500 g. Culture supernatants were thawed and spun down vertically at 4 °C for 20 min at 2000*g* and then centrifuged horizontally at 100,000*g* for 75 min. Finally, the supernatant was discarded and the exosome pellet was re-suspended in 1 mL of PBS. The isolated EVs were marked with PKH26 (Red Fluorescent Cell Linker Kits MINI26; Sigma-Aldrich Co., St Louis, MO, USA) for 5 min at room temperature in dark room and blocked with fetal bovine serum, according to manufacturer’s instructions. Unincorporated labels were removed by exosomes centrifugation at 4 °C for 75 min at 100,000*g*. Finally, exosomes were washed with DPBS and subjected to additional centrifugations.

### 2.7. Seeding Exosomes on the Scaffolds and Cell Culture in Proximity of Exosomes-Enriched-Scaffolds

Exosomes were resuspended into 500 µL and seeded drop by drop onto the surface of the scaffolds at a density of 5 × 10^10^ cm^2^ (Figure 1a) and the scaffolds enriched with exosomes were placed in a humid incubator for 24 h. Then, the exosomes-enriched scaffolds with the exosomes were placed in the plastic well and then cells were seeded inside the same plastic well (Figure 1b) at a density of 5 × 10^5^ and cultured in dedicated growth medium up to rich confluence. The effect of the exosomes released by the scaffolds on stem cells commitment has been evaluated by means of gene expression (Figure 1c).

### 2.8. Osteogenic Commitment by Real-Time Polymerase Chain Reaction (RT-PCR)

RNA extraction was made using total RNA Purification Plus Kit (Norgen Biotek Corporation, Thorold, ON, Canada), following the manufacture instructions and stored at −80 °C. Total RNA obtained from the specimens was reverse-transcribed with SensiFAST™ cDNA Synthesis kit (Bioline GmbH, Konstanz, Germany) in a LifePro Thermal Cycler (BioerTechnology, Hangzhou, China) following manufacturer instructions; 10 min annealing at 25 °C, 45 min reverse transcription at 42 °C, and 5 min inactivation at 85 °C.

The obtained cDNA specimens were stored at −20 °C until their use. Gene expression of collagen type I, runx, osteonectin, and osteocalcin as common osteogenesis markers, was investigated by RT-PCR. Human primers were previously selected for each target gene with Primer 3 software (Thermo-Fisher Scientific, Waltham, MA, USA).

Then, RT-PCR was performed using the selected primers at 400 nM concentration and a kit (SensiFAST™ SYBR No-ROX, Bioline GmbH, Luckenwalde, Germany) on a cycler (Rotor-Gene 3000, Corbett Research, Sydney, Australia).

The used thermal cycling conditions were: 2 min denaturation at 95 °C, followed by 40 cycles of denaturation for 5 s at 95 °C, annealing for 10 s at 60 °C, and elongation for 20 s at 72 °C. Data analysis was performed by 2ΔΔCt method.

The threshold cycle (Ct) values (i.e., the fractional cycle number at which the amount of amplified target reaches a determined threshold) of target genes were normalized to a housekeeping gene (transferrin receptor 1).

The relative gene expression between control group (hAD-MSCs on plastic monolayer) and test group (hAD-MSCs cultured in presence of exosomes released from the scaffolds) was evaluated. The results were reported as fold regulation of target genes in test group compared with hAD-MSCs in monolayer; fold regulation indicate increased (values > 2) or decreased (values < −2) gene expression, while values comprised between −2 and 2 indicate indifferentially expressed genes.

### 2.9. Nanoparticle Characterization of Exosomes Isolated from hAD-MSCs

The size and concentration of hAD-MSCs-EVs were analyzed using a high-resolution system (NanoSight NS300, Malvern Instruments, Malvern, UK) configured with scientific camera (CMOS, Mightex Systems) and blue 488 nm laser. Collected EVs were diluted in 1 mL of DPBS and analyzed using a software (NTA Version 3.2). Each sample from different isolations was recorded three times for 60 s at 23 °C temperature. Three replicable histograms were created, the values were then averaged.

### 2.10. Transmission Electron Microscopy (TEM) Observation of Released Exosomes

The exosomes were fixed overnight at 4 °C in 2.5% glutaraldehyde/0.1 M sodium cacodylate buffer solution. Then samples were treated with 1% OsO_4_/0.1 M sodium cacodylate buffer, dehydrated through increasing the concentration of ethanol solutions and then fixed in epoxy resin (EPON™, Hexion, Houston, TX, USA). Ultrathin sections (ultramicrotome, LKB, Stockholm, Sweden) were prepared, stained with heavy metal solutions (1% uranyl acetate and 1% lead citrate), and analyzed by TEM (Tecnai G12, FEI Company, Hillsboro, OR, USA) at acceleration voltage 100 kV. The image acquisition was performed by a video camera (Tietz, Tietz Video and Image Processing Systems GmbH, Gauting, Germany) and an imaging software (TIA, FEI Company, Hillsboro, OR, USA).

### 2.11. Immunofluorescence Staining of Exosomes

Exosomes were fixed for 10 min in 4% paraformaldehyde (Sigma-Aldrich, St. Louis, MO, USA), washed three times and incubated for 1 h at room temperature in PBS solution with 3% bovine serum albumin (BSA, Sigma-Aldrich). Finally, cells were incubated at 4 °C overnight with primary mouse antibodies anti-human CD 81, CD63, CD 9 (Thermo Fisher Scientific, MA, USA) and then with the fluorescent secondary goat antibodies anti-rabbit (Alexa Fluor 555 dye, Thermo Fisher Scientific, MA, USA). Immunofluorescence images were acquired by a microscope (Upright ECLIPSE Ni Microscope, Nikon, Minato, Tokyo, Japan).

## 3. Results

### 3.1. PLA-10CaSi-10DCPD Scaffolds

ESEM analysis at 500× on fresh samples displayed a surface characterized by a thick structure with limited, but regular micropores. Pores were mostly circular shaped, the diameter ranged from 20 to 100 μm (Figure 2).

Mineral granules from CaSi and DCPD powders (mean size of 1–3 µm) were present along all the surface, in some cases, they were agglomerated in larger granules (approx. 20–30 µm).

EDX microanalysis on samples revealed the constitutional elemental peaks of PLA, namely oxygen (O), carbon (C), and reflexes of Si, Ca, and P from the incorporated CaSi and DCPD granules. The mean value of the surface porosity at 100×, 500×, and 1000× magnifications, were 25.20 (range 24.73–28.65), 32.01 (range 29.19–34.85), and 23.68 (range 19.58–34.68), respectively (Figure 3).

Samples immersed in HBSS for 28 days revealed a mineral layer on the surface, with an uniform distribution evident at 100× magnifications, indicating that the mineral layer filled the scaffolds pores in numerous areas. In a few, some porosities were observable.

ESEM observations at 500× and 1000× magnifications on one area revealed a uniform distribution of the mineral filler, revealing in some cases a partially degraded matrix (Figure 2).

The high variability found at high magnifications and the overall decrease of surface porosity may be attributable to calcium phosphate nucleation, which was marked on these scaffolds.

EDX confirmed the elemental constitution of the mineral layer. The analyses revealed the elemental peaks from the HBSS solution, a marked increase of P, the intensity reduction of PLA structural peaks C and O and increase of elements belonging to the CaSi granules, the latter may indicate the progressive dissolution of the polymeric matrix that exposed additional mineral filler.

Surface porosity decreased when compared to fresh samples; the mean values at 100× were 20.61 (range 18.54–23.78), at 500× were 27.66 (range 23.06–32.05), and at 1000× was 22.18 (range 18.03–28.43) (Figure 3).

### 3.2. PLA-5CaSi-5DCPD Scaffolds

ESEM analysis performed at 500× magnification evidenced a different surface compared to PLA-10CaSi-10DCPD. A homogeneous layer from CaSi and DCPD incorporated powders partially occludes the matrix porosities, the shape of the pores was mostly circular (size approx. 20 μm). At 1000× magnifications, the mineral powder may be better identified, revealing small electron dense granules (< 10 μm), which resulted agglomerated in larger cuboidal deposits (Figure 4).

EDX revealed C and O both ascribable to the polymer, and Ca, Si, and P peaks (from the incorporated mineral fillers).

Surface porosity was markedly increased when compared to PLA-10-CaSi-10DCPD, the mean values where 49.02 at 100× (range 46.15–52.29), 45.14 at 500× (37.53–50.12), 46.29 at 1000× (range 42.22–48.13) (Figure 5).

After HBSS soaking for 28 days, an irregular surface was detected. ESEM at 100× revealed a more compact mineral layer, which covers the scaffolds pores. The higher surface of the PLA structure was found degraded in some areas, identified at high magnifications, where the CaP layer was not present. In this areas, larger irregular pores were detected (approx. 20–100 µm) (Figure 4). EDX microanalysis of the entire area showed a stability of Ca and P and Si, which may suggest exposure of CaSi/DCPD granules and some CaP nucleation on the degraded PLA structure. Surface porosity, decreased from the fresh samples, the mean values where 39.02 at 100× (range 36.07–42.08), 42.24% at 500× (35.09–48.06), 40.08 at 1000× (range 36.05–43.03) (Figure 5).

### 3.3. In Vitro Biocompatibility of Scaffolds

Biocompatibility of the scaffolds has been evaluated by MTT test of cells seeded onto the scaffolds. The MTT assay proved that hAD-MSCs were able to proliferate on both scaffold formulations (Figure 6). The OD values recorded for cells loaded on PLA-10CaSi-10DCPD samples were found to be significantly (*p* < 0.05) higher to those observed for the PLA-5CaSi-5DCPD. In any case, the results confirm that both samples supported cells growth. Rectangular bottom plastic flasks manufactured from optically clear polystyrene, sterilized by gamma irradiation and PLA are used as control.

### 3.4. Analysis of EVs Isolated from hAD-MSCs

EVs derived from hAD-MSCs were a homogeneous population of exosomes/microvesicles (average peak 150.3 ± 8.7 nm) (Figure 7a).

The majority of these particles revealed a cup or round shape morphology, also expressing the typical exosome surface markers, such as CD9, CD63, and CD81 (Figure 7b–d).

TEM and immunohistochemistry were used to characterize the purified hAD-MSCs-derived exosomes (Figure 7e).

### 3.5. Exosome Adhesion on Scaffolds and Their Release

Scaffolds enriched with red labeled exosomes have been inserted into a multiwell plate. The ability of the exosomes to be entrapped by the scaffolds surface is well evident as red-labeled vesicles are present on the surfaces of both experimental scaffolds (Figure 8).

The hAD-MSCs internalization of the exosomes released by the scaffolds was detected from 3 to 24 h of culture, using cells seeded on a multiwell plastic surface containing the scaffolds enriched with the exosomes.

The results showed that the red-labeled exosomes are present inside the cytoplasmatic region of hAD-MSCs (Figure 9) indicating that hAD-MSCs were able to uptake the exosomes.

### 3.6. Effect of Exosomes on Osteogenic Commitment of hAD-MSCs

The ability of the scaffolds to release active and pro-osteogenic exosomes has been evaluated by means of gene expression. Cells were seeded in monolayer condition near the scaffolds.

PLA-5CaSi-5DCPD plus exosomes and PLA-10CaSi-10DCPD plus exosomes have been tested; the control was represented by the scaffolds without exosomes to evaluate the ability of the scaffolds to release some osteogenic soluble factor.

Gene expression of collagen type I, osteopontin, osteonectin, osteocalcin and runx was higher for the mineral-doped scaffolds than for PLA control scaffold (Figure 10). The PLA-10CaSi-10DCPD enriched with exosomes showed the highest ability to stimulate the expression of osteogenic markers. The osteogenic commitment is also stimulated by PLA-10CaSi-10DCPD, PLA-5CaSi-5DCPD, and PLA-5CaSi-5DCPD enriched with exosomes compared with PLA control (Figure 10).

The data indicated that for all the five scaffolds the osteogenic commitment of cells is present, that the mineral fillers CaSi-DCPD could improve this process and that the higher level of commitment is present when exosomes from the samples are released in the medium.

## 4. Discussion

Biomaterials able to release biologically active ions have been considered a promising strategy for bone regeneration [43,44,45,56,57,58,59,60]. It has been reported that chemical signals from biomaterials can promote the osteogenic differentiation of mesenchymal stem cells [47]. More importantly, the chemical or structural signals of biomaterials could influence the cell–cell communication between neighboring cells, consequently leading to significantly accelerated tissue regeneration [47].

Exosomes have recently emerged as potential mediators with paracrine capacity in cell-cell communication [61]. Exosomes could deliver various informational molecules, including functional proteins, mRNAs, and microRNAs, and consequently regulate the behaviors of adjacent or distant cells. The functional properties of exosomes have been widely recognized to depend mainly on their molecular components, which are representative of their original producing cells. Recently it has been demonstrated that MSC-derived exosomes have therapeutic potential in various disease models in the absence of MSCs [8,9,10] and also in oral and maxillo-facial experimental models [62,63,64,65,66].

In this study, scaffolds composed of a PLA and incorporating different percentages of bioactive materials were enriched with exosomes secreted from hAD-MSCs. The collected exosomes resulted a homogeneous population of cup- or round-shaped vesicles having 110–180 nm diameter and expressing the typical exosome surface markers, such as CD9, CD63, and CD81 [67].

Our results indicated all PLA-based formulations (mineral-doped and pure PLA) were able to uptake and release exosomes and that exosomes released from the different PLA-based scaffolds were able to significantly enhance the osteogenic commitment of hAD-MSCs. These data are consistent with a recent study, where 3D-printed PLA-based scaffold enriched with exosome increased hGMSCs gene expression [63].

Our data showed that exosomes interacted with hAD-MSCs through membrane to membrane fusion, delivering their content into the targeted cells and triggering their osteogenic commitment, as shown in Figure 9. Indeed, hAD-MSCs incubated in the presence of all the experimental exosomes-enriched scaffolds showed increased expression of collagen type I, runx, osteonectin, osteocalcin all biochemical markers for osteogenesis. The data suggest that exosome may improve the cell gene expression also in the PLA-based scaffold free from mineral components. Interestingly, PLA-10CaSi-10DCPD i.e., the formulation doped with the uppermost amount of mineral fillers showed the highest gene expression in either the presence or absence of exosomes.

In this study, a PLA matrix was selected for its biological properties, making it a suitable material for several biomedical applications and for its green production as it is a bio-derived polymer produced by bacterial fermentation using renewable resources [29,68]. The PLA used in our formulations has been prepared by the company through ring opening lactide polymerization with no use of solvents [29]. After the complete polymerization, any remaining lactide monomer has been removed and recycled within this process [30,31]. In the present study, the commercial PLA has been further purified through dissolution in CHCl_3_ and chemical reprecipitation in cold MeOH, as confirmed by previous NMR experiments (data not shown) [52]. The scaffolds have been produced by TIPS, a procedure allowing the production of highly porous scaffolds also with high amount of fillers, having low production cost and easy processability [52]. Other traditional techniques such as solvent cast method, or additive manufacturing procedures would not been able to obtain high porosity values and a homogeneous dispersion of CaSi and DCPD, for the different density of PLA matrix and the mineral filler [52]. Similarly, other techniques produced materials with reduced mechanical properties; a study on PLA-based scaffolds created using increasing percentages of porogen concentration (from 5% to 30%) demonstrated a high bulk porosity (approx. 80%) with a significant decrease in the mechanical properties (compressive strength and compressive modulus) [69].

In the present study, CaSi filler have been selected as bioactive doping materials as currently used in endodontic field for their positive bio-interaction with bone tissues (i.e. periapical bone and pulp regeneration) [38,46]. 

Because of their chemical [70,71] and biological [46,56,72] properties, calcium phosphate components were introduced in the scaffolds composition to obtain Ca and P releasing materials able to provide positive epigenetic chemical signals to the cells involved in regenerative processes. DCPD was chosen as the filling material as it improves and accelerates the apatite nucleation of CaSi-based materials providing an additional phosphate source in the early phases after immersion because of its progressive dissolution [42,73]. DCPD is a material used to prepare osteotransductive bone cements [74] possessing very attractive properties as it converts into apatite after contact with water and as quite soluble in water due to the presence of structural water; DCPD is the most soluble calcium orthophosphate salt at pH > 8.2 [75].

Our results demonstrated that mineral-doped PLA-5CaSi-5DCPD and PLA-10CaSi-10DCPD scaffolds increase cell gene expression with respect to PLA scaffold free from the mineral components. In addition, hAD-MSCs cultured in the presence of PLA-10CaSi-10DCPD showed the highest expression of osteogenic markers, mainly in the presence of exosome-enriched PLA-10CaSi-10DCPD demonstrating that the bioactivity of scaffold components plays a positive role in gene expression.

The presence of exosomes improves the osteogenic commitment of hAD-MSCs for all the experimental scaffolds, including the control scaffold (pure PLA). This data is consistent with a recent study, where 3D-printed PLA-based scaffold enriched with exosome increased the gene expression of human gingival mesenchymal stem cells [63].

The structural porosities of the experimental scaffolds have been deeply studied in previous investigations [45,52]. Micro-CT analysis of these scaffolds [52] revealed a highly porous (approx. 90%) and highly interconnected structure (percentage of open pores > 99%), and a uniform structure with homogeneous dispersion of CaSi and DCPD. The homogeneous dispersion of the mineral filler was previously correlated to the scaffolds biointeractivity (release of Ca^++^ and OH^−^ ions) [52]. Moreover, PLA-10CaSi-10DCPD (the scaffold with the higher mineral filler content) revealed the higher structural wall thickness (thickness of the walls among interconnected pores) [45].

When considering bioactive materials, their dynamic modification in simulated body fluid need to be investigated to mimic/simulate the material changes and the surface modifications such as polymer degradation, filler dissolution, and calcium phosphate nucleation.

Our results demonstrated that PLA matrix only partially degraded, that a mineral calcium phosphate layer formed on the surface partially filling the surface pores, which remained open to potential blood influx into the scaffold core. ESEM-EDX morphometric analysis of the designed scaffolds surface showed a variable percentage of surface porosity, decreasing when a higher content of mineral filler was added. Surface pores were mostly circular-shaped in both formulations and their distribution was uniform in all the scaffold structure. Similarly, the incorporated CaSi and DCPD fillers where widespread throughout all the structure, with some agglomeration revealed by ESEM investigation. These data are consistent with previous results obtained from experimental PLA mineral-doped scaffolds [52] and also with poly-e-caprolactone mineral-doped scaffolds [76].

The release of biologically active ions and the formation of a calcium phosphate layer on the surface may affect the behavior and gene expression of hAD-MSCs and trigger their osteogenic commitment. Indeed, calcium and silicon ions released from CaSi demonstrated to stimulate mineralizing cells proliferation [43,44,45,47,56,57,58,59,60] enhancing the expression of several genes, such as pro-osteogenic genes (as osteocalcin, bone sialoprotein and alkaline phosphatase) [42,43,44,45,77,78,79] and pro-angiogenic genes [60]. PLA-based CaSi-DCPD-doped scaffolds revealed the ability to support the growth of vascular wall mesenchymal stem cells, stimulating the gene expression of several pro-angiogenic markers, including platelet-derived growth factor receptor-β (PDGFR-β), alpha smooth muscle actin (α-SMA), neuron-glial antigen 2 (NG2), and CD 90 [60].

Stem cell-derived exosome therapy for bone healing is drawing increasing attention in the biomedical field [80]. Today, exosomes ranging in size from 50–120 nm, provide researchers a novel way to stimulate bone regeneration with safer approach and powerful pro-osteogenic ability. This strategy helps to promote proliferation and migration of cells, as well as osteogenesis and angiogenesis, in the process of bone formation.

Although the exact mechanisms remain elusive, exosome miRNAs seem to play vital roles. Exosomes directly regulate and guide MSCs into the osteoblastic lineage [79]. MSC-derived exosomes can be used as biomimetic tools to induce naïve stem cells to a osteogenic linage.

Profiling data for the MSC-derived exosome have revealed that nine miRNAs (let-7a, miR-199b, miR-218, miR-148a, miR-135b, miR-203, miR-219, miR-299-5p, and miR-302b) are up-regulated and four miRNAs (miR-221, miR-155, miR-885-5p, miR-181a and miR-320c) are down-regulated during the process of MSC osteoblastic differentiation [79]. All of these miRNAs have roles in osteoblast function and activity.

In light of these considerations, in the present study we proposed the use of exosomes to functionalize biomaterials obtaining a cell-free system composed by exosomes-enriched scaffold to improve the osteogenic ability of the scaffolds/biomaterials without immunogenic effect. Compared to direct cell seeding, exosomes-enriched scaffold strategy may provide several advantages, including high stability, intrinsic homing effect, and low immunogenicity [81,82,83].

Innovatively, the present work demonstrated that the experimental mineral-doped porous scaffolds were able to uptake and hold exosome vesicles on their surface. Moreover, the exosome-enriched scaffolds can release the exosomes in functional form into the medium and, when placed in proximity of cultured hAD-MSCs, the exosome-enriched scaffolds induce cell commitment toward osteogenic lineage.

The proposed model can be considered as a novel strategy to improve, once implanted in vivo, a potential pro-osteogenic effect on MSCs present on the healing environment.

Additional studies are planned to assess the long-term effects of exosome-enriched scaffolds and whether this strategy could be translated in clinical bone regenerative applications; in particular the scaffold doped with the higher amount of mineral fillers (PLA-10CaSi-10DCPD) deserve additional animal model investigations and further in vivo investigations to demonstrate its great potential for bone regenerative purposes.

## 5. Conclusions

The study showed that:The experimental PLA-based scaffolds are able to adsorb, keep on their surface, and release exosomes secreted by hAD-MSCs.The exosome-enriched scaffolds have enhanced ability to trigger the osteogenic commitment of hAD-MSCs improving their osteogenic properties.Mineral-doped scaffolds, in particular the formulation with the highest amount of mineral fillers (PLA-10CaSi-10DCPD), showed a great potential in regenerative bone healing by stimulating the osteogenic commitment of hAD-MSCs.

## Figures and Tables

**Figure 1 nanomaterials-10-00432-f001:**
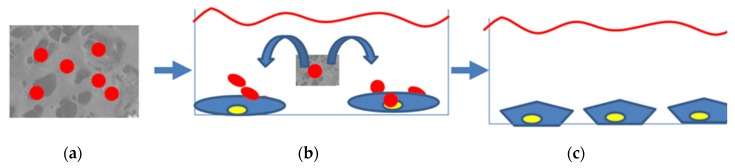
Experimental design direct to analyze the effect of exosomes enriched scaffolds on human adipose mesenchymal stem cells (hAD-MSCs) commitment. (**a**) Scaffold enriched with exosomes; (**b**) exosomes released in the culture media can be uptaken by cells and trigger their commitment; (**c**) hAD-MSCs commitment into polygonal shaped osteoblastic feature induced by exosomes.

**Figure 2 nanomaterials-10-00432-f002:**
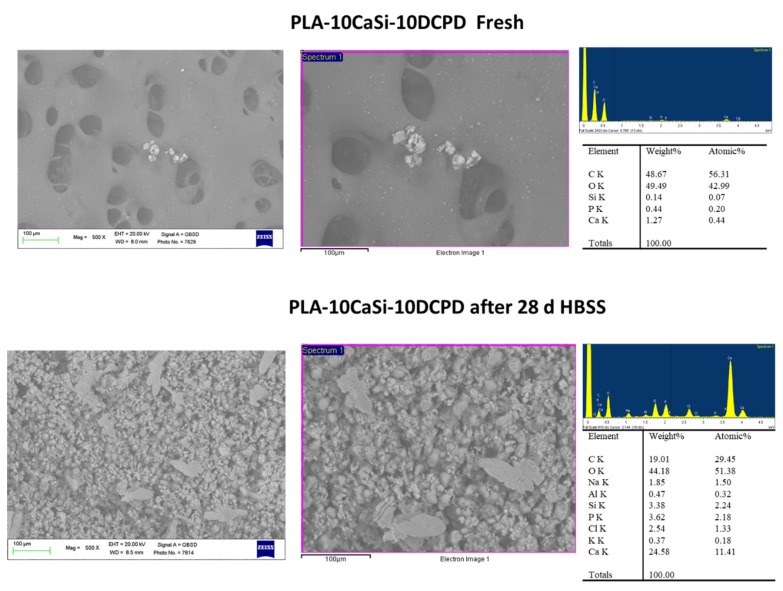
Surface micromorphology of PLA-10CaSi-10DCPD before and after 28 days immersion in simulated body fluid (500×–1000× magnification); pores of different size (diameter range between 20 and 100 µm) were observed: CaSi and dicalcium phosphate dihydrate (DCPD) microgranules are diffused onto all the structure. In some areas, agglomeration of these granules were also observed. Samples soaked in HBSS for 28 days revealed a well-evident electron dense mineral layer on the materials with pores of approx. 100 µm diameter filled by this layer. Energy dispersive X-ray (EDX) microanalysis on samples not immersed in HBSS showed the polylactic acid (PLA) structural peaks (C and O) and low traces of mineral filler constitutional peaks (Ca, Si, and P); EDX revealed after 28-day immersion in simulated body fluid a consistent decrease of C (from PLA matrix), and a significant elevation of Ca, Si, and P.

**Figure 3 nanomaterials-10-00432-f003:**
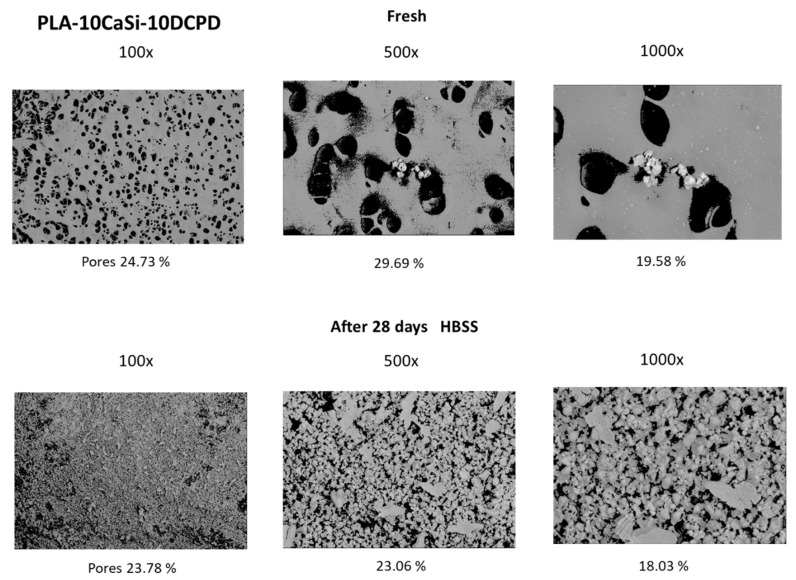
Mineral-doped surface micromorphology by environmental scanning electron microscope (ESEM) of PLA-10CaSi-10DPCD scaffold. Differences in surface porosity were observed when considering sample immersed for 28 days in simulated body fluids, attributable to calcium phosphate mineral layer deposition which partially covers the polymeric porous matrix.

**Figure 4 nanomaterials-10-00432-f004:**
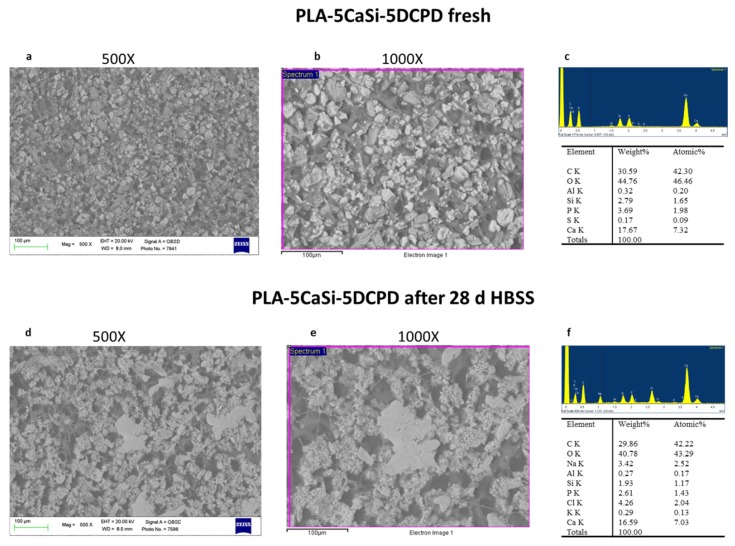
SEM-EDX microanalysis before and after the immersion in HBSS of PLA-5CaSi-5DCPD scaffold. Fresh scaffolds revealed a homogeneous mineral layer, partially occluding the superficial scaffold pores (size of approx. 20 μm). EDX confirmed the presence of CaSi and DCPD mineral powders. After 28 days immersion in HBSS, the scaffold surface resulted more irregular: in the majority of cases, the surface area was completely covered by a compact mineral later, while in few cases, the surface revealed larger pores (ranging from 20–100 µm), attributable by PLA initial degradation. EDX revealed a moderate increase of Ca and P (from CaP layer) and a slight increase of Si (from the matrix degradation).

**Figure 5 nanomaterials-10-00432-f005:**
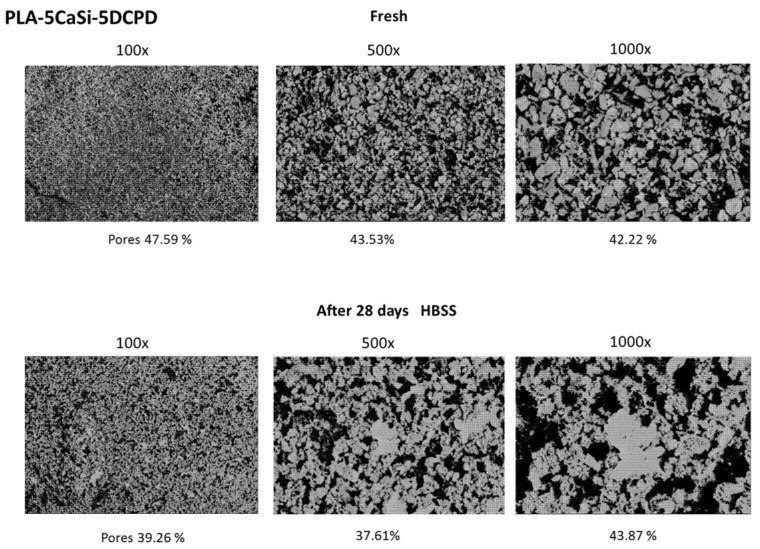
ESEM surface micromorphology of PLA-5CaSi-5DCPD scaffold. An overall decrease of surface pores was observed at all magnifications, attributable to the mineral layer deposition upon immersion in HBSS.

**Figure 6 nanomaterials-10-00432-f006:**
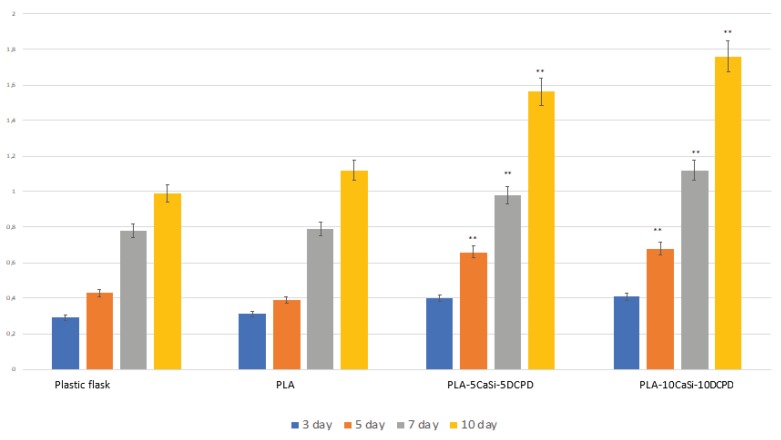
MTT assay of hAD-MSCs cultured for 10 days in plastic flasks without scaffolds, on PLA scaffolds (both used as control) without mineral components, and on PLA-5CaSi-5 DCPD or PLA-10CaSi-10DCPD scaffolds. T-tests were used to determine the significant differences (*p* < 0.05). * *p* < 0.05; ** *p* < 0.01; *** *p* < 0.001.

**Figure 7 nanomaterials-10-00432-f007:**
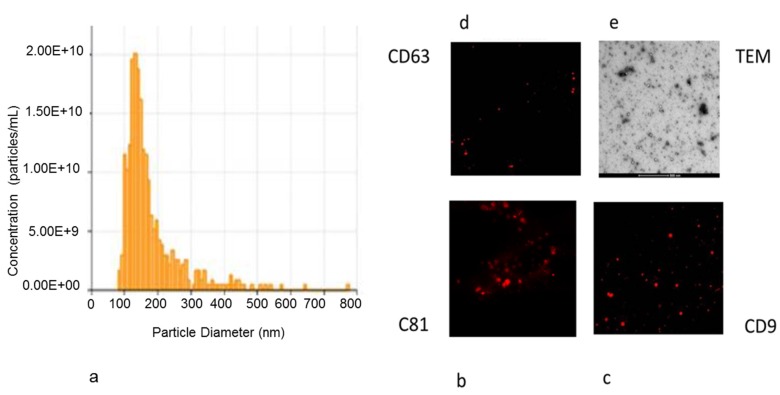
Characterization of exosomes derived from hAD-MSCs. (**a**) Size and concentration (by NanoSight analysis) of exosomes secreted by hAD-MSCs is reported in the graph. Nanoparticle characterization showed two populations of exosomes with peak diameter of 110 and 180 nm, respectively. (**b**–**d**) Immunohistochemistry analysis of the exosome surface markers CD9, CD 63, CD 81. (**e**) Transmission electron microscopy (TEM) image of exosomes released by hAD-MSCs.

**Figure 8 nanomaterials-10-00432-f008:**
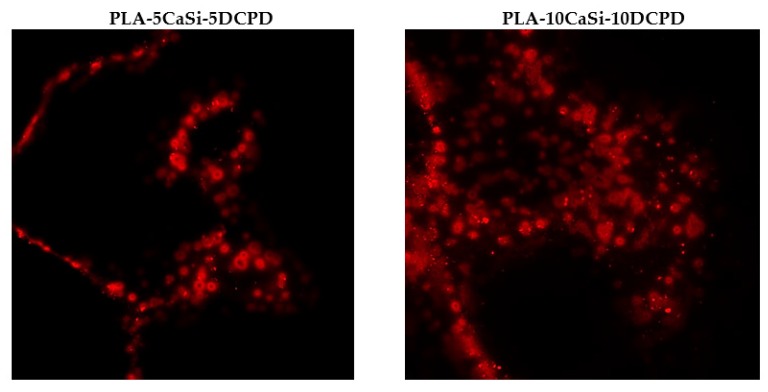
Red labeled exosomes on PLA-5CaSi-5DCPD (**left**) and PLA-10CaSi-10DCPD (**right**) scaffolds.

**Figure 9 nanomaterials-10-00432-f009:**
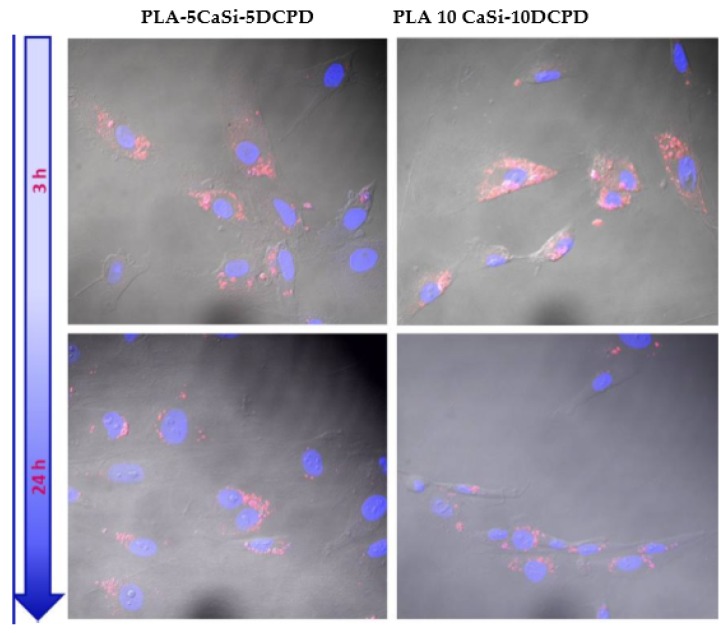
Internalization of exosomes released from the scaffolds inside hAD-MSCs.

**Figure 10 nanomaterials-10-00432-f010:**
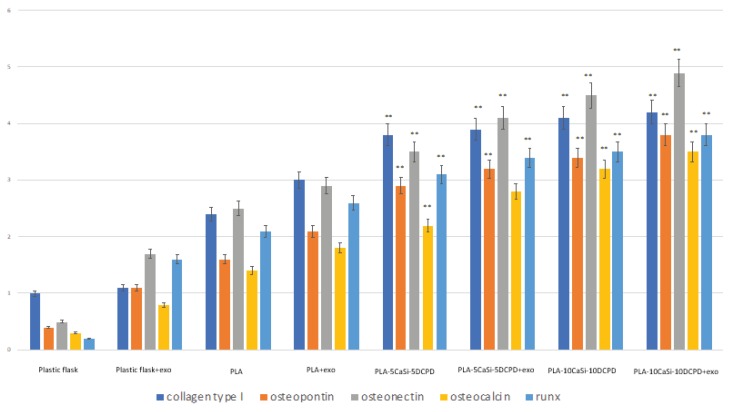
Gene expression of hAD-MSCs cultured on monolayer in plastic flasks, in plastic flasks with exosomes, in plastic flasks in presence of PLA (control), in presence of PLA-5CaSi-5DCPD or PLA-10CaSi-10DCPD scaffolds, and in presence of PLA-5CaSi-5DCPD scaffolds enriched with exosomes or PLA-10CaSi-10DCPD scaffolds enriched with exosomes. *T*-tests were used to determine significant differences (*p* < 0.05). * *p* < 0.05; ** *p* < 0.01; *** *p* < 0.001 in comparison with PLA control. No statistical significance for plastic flask and plastic flask + exo, is present.

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
