# Peer review of "Mineral-Doped Poly(L-lactide) Acid Scaffolds Enriched with Exosomes Improve Osteogenic Commitment of Human Adipose-Derived Mesenchymal Stem Cells"

_nanomaterials, 2020, doi:10.3390/nano10030432_

Round 1

Reviewer 1 Report

The reviewed work presents the results of studies on the mesenchymal stem cells culture carried out on scaffolds made of poly (L-lactide) and enriched by exosomes. Extracellular vehicles - exosomes derived from stem and progenitor cells may have therapeutic effects comparable to their parental cells and are considered promising agents for the treatment of a variety of diseases. Similar studies have been conducted before, but it is rather unlikely to be argued that the manuscript does not contain scientific novelty.

The main goal of this presented study was to investigate the effect of cell culture conditions on osteogenic commitment of human adipose mesenchymal stem cells seeded on PLA scaffolds contained calcium silicate, di-calcium phosphate and enriched with exosomes vesicles. In this type of research, it is very important to choose the right scaffold - material and method of manufacturing, for obtaining optimal properties of the final matrix. In the presented investigations, the authors used only two types of scaffolds differing only in their mineral content, which makes it difficult to obtain correct, accurate conclusions. In this case, the only differentiating factor is the increase in the amount of these minerals. Unfortunately, they chose the unsuitable polymer material, which can also negatively effect on the final results. The presented work can be accepted for publication, but it requires rewording as well as many explanations and additions. Because the aim of the research and the final result are very unclear, authors should to present the clarified conclusions of the research, show what exactly is the main achievement compared to the research conducted so far?

The main doubts are:

- why PLA 4060D, Natureworks LLC was used in scaffold forming, this polymer is not intended for biomedical applications, but only for forming food packaging. It is a material containing processing enhancers that can affect cell proliferation. Polymers intended for biomedical applications, for forming implants or drug carriers must be used in this type of research.

- line 76-77 - high molecular weight poly (L-lactide) for medical and packaging purposes is obtained from L-lactide by ROP (this lactide is obtained from lactic acid), this procedure allows obtaining high molecular weight product with high chemical purity.

- line 99 - in no case can it be possible to remove all traces of the initiator by precipitating a polylactide solution, especially in the case amphoteric tin compounds (please read the work of Prof. M. Vert on this issue). Therefore, the type of initiator used during polymerization is of great importance in the biomedical applications of such polymers.

- line 110 - the morphology of the obtained and used scaffolds should be described in more detail (content of pore, their homogeneity in the entire volume or  their gradient distribution, pore size distribution, what percentage of pore was the nature of open pores, etc.

- line 140 - please describe in detail the manner in which exosomes were seeded on the scaffolds. Was the cell culture carried out only on the surface of the scaffold?

- line 306 - based on only in vitro cell growth studies, it is not possible to determine the biocompatibility of the material, this requires a much wider range of studies.

- Figure 7 - in this type of research the results obtained with the participation of scaffold without minerals as well as on a neutral surface (e.g. TCPS) should be demonstrated as a reference test

- Figure 10 - here as a reference should be presented the results of tests on a neutral surface (e.g. TCPS)

- line 364 - the authors claim that the PLA matrix due to the rate of degradation is the most suitable in bone tissue engineering, because PLGA degrades too quickly. This is not true, as evidenced by the huge amount of work published so far, as well as the results of presented clinical and preclinical studies.

- lines 376-380 - these are facts described and known, this information is not suitable for Conclusions part , but rather for the Introduction part.

The choice of material was unfortunate. In order for the work to make sense, it needs to be supplemented and rewritten, so as to emphasize the importance of exosomes, as well as the impact of mineral additives in the matrix. In the present form, the work is not suitable for publication, despite professionally conducted experiments. The authors clearly ignored the issues related to the proper selection of material and the method of forming optimal cell's scaffolds, which is extremely important if we want to obtain satisfactory results of conducted cultures and be able to compare them with the results of other researchers.

Author Response

To the editorial Board of Nanomaterials,

The manuscript has been now deeply reviewed and the issues raised by the referees resolved. All the changes have been highlighted in yellow.

REFEREE 1

The reviewed work presents the results of studies on the mesenchymal stem cells culture carried out on scaffolds made of poly (L-lactide) and enriched by exosomes. Extracellular vehicles - exosomes derived from stem and progenitor cells may have therapeutic effects comparable to their parental cells and are considered promising agents for the treatment of a variety of diseases. Similar studies have been conducted before, but it is rather unlikely to be argued that the manuscript does not contain scientific novelty.

The main goal of this presented study was to investigate the effect of cell culture conditions on osteogenic commitment of human adipose mesenchymal stem cells seeded on PLA scaffolds contained calcium silicate, di-calcium phosphate and enriched with exosomes vesicles. In this type of research, it is very important to choose the right scaffold - material and method of manufacturing, for obtaining optimal properties of the final matrix. In the presented investigations, the authors used only two types of scaffolds differing only in their mineral content, which makes it difficult to obtain correct, accurate conclusions. In this case, the only differentiating factor is the increase in the amount of these minerals. Unfortunately, they chose the unsuitable polymer material, which can also negatively effect on the final results. The presented work can be accepted for publication, but it requires rewording as well as many explanations and additions. Because the aim of the research and the final result are very unclear, authors should to present the clarified conclusions of the research, show what exactly is the main achievement compared to the research conducted so far?

The main doubts are:

- why PLA 4060D, Natureworks LLC was used in scaffold forming, this polymer is not intended for biomedical applications, but only for forming food packaging. It is a material containing processing enhancers that can affect cell proliferation. Polymers intended for biomedical applications, for forming implants or drug carriers must be used in this type of research.

The selection of PLA Ingeo 4060D is justified by the easy availability of this commercial PLA grade.The presented study does not include in vivo testing and small animal implants and the cell viability test (MTT assay) at 3-5-7 and 10 days of contact between scaffolds and cells shows no toxicity and hAD-MSCs were able to proliferate. Similar scaffolds prepared by TIPS with PLA Ingeo 4060D were previously tested (Polylactic acid-based porous scaffolds doped with calcium silicate and dicalcium phosphate dihydrate designed for biomedical application. (Gandolfi et al. 2018) in contact with mouse embryo fibroblasts balb/3 T3 clone A31 cell line (CCL-163) and no citotoxicity was oberved also in this case.  In  another recent paper (Tatullo et al. 2019), similar PLA based scaffolds were tested in presence of human periapical cyst mesenchymal stem cells, demonstrating the suitability of the scaffolds to be colonized by this MSC population.Furthermore, biodegradation tests which are particularly affected by catalyst residues potentially released from the polymer matrix, are not included in the presented work.

Moreover, we did not found traces of Sn (possible residue of stannous octoate used as ROP) in the scaffolds by EDX analyses nor by micro Raman (micro Raman has been reported in the previous investigation on the chemical physical characterization of these experimental scaffolds) (Gandolfi 2018). In addition, Safety data sheet of PLA 4060D (https://www.natureworksllc.com/~/media/Files/NatureWorks/Technical-Documents/Safety-Data-Sheets/NA-ENG/SDS_NatureWorks_Ingeo-4060D_pdf.pdf) reports no carcinogenic or mutagenic effects and no acute toxicity in specific target organs.

Gandolfi MG, Zamparini F, Degli Esposti M, Chiellini F, Aparicio C, Fava F, et al. Polylactic acid-based porous scaffolds doped with calcium silicate and dicalcium phosphate dihydrate designed for biomedical application. Mater Sci Eng C. 2018;82:163-181.

- line 76-77 - high molecular weight poly (L-lactide) for medical and packaging purposes is obtained from L-lactide by ROP (this lactide is obtained from lactic acid), this procedure allows obtaining high molecular weight product with high chemical purity.

The sentence has been modified. Further information on PLA industrial production has been added in the discussion section.

- line 99 - in no case can it be possible to remove all traces of the initiator by precipitating a polylactide solution, especially in the case amphoteric tin compounds (please read the work of Prof. M. Vert on this issue). Therefore, the type of initiator used during polymerization is of great importance in the biomedical applications of such polymers.

Quantitative Nuclear magnetic resonance (qNMR) experiments performed on reprecipitated PLA used for the preparation of the scaffolds in order to assess the high purity of the polymer, demonstrated that no impurities and contaminants can be detected. This means that if unwanted substances are contained, their concentration is extremel. In addition no traces of Sn (possible residue of stannous octoate used for ROP) in the fresh scaffolds by EDX analyses in the present study, or by micro Raman investigation previously conducted [Gandolfi 2018].

- line 110 - the morphology of the obtained and used scaffolds should be described in more detail (content of pore, their homogeneity in the entire volume or  their gradient distribution, pore size distribution, what percentage of pore was the nature of open pores, etc.

Due to the easily compressibility of highly porous polymer scaffolds, reliable results concerning porosity and pore size distribution cannot be obtained with mercury intrusion porosimetry.

In (Gandolfi et al. 2018), scaffolds prepared with the same compositions and technique were studied in terms of bulk open porosity according to the standard procedure UNI EN 1936. Open porosity of PLA-based scaffolds obtained by TIPS are approximately 92% and type and amount (arriving at 20%) of filler do not affect this value.

In (Tatullo et al. 2019), PLA based scaffolds have been investigated through MicroCT analysis.  Additional information on scaffolds micro morphology, pore distribution interconnected porosities and structural wall thickness have been now included in the discussion section.

In the present study authors focused on surface modifications before and after immersion in HBSS as surface topography is particularly relevant for cells seeding, growth and proliferations.

Gandolfi MG, Zamparini F, Degli Esposti M, Chiellini F, Aparicio C, Fava F, et al. Polylactic acid-based porous scaffolds doped with calcium silicate and dicalcium phosphate dihydrate designed for biomedical application. Mater Sci Eng C. 2018;82:163-181.

Tatullo M, Spagnuolo G, Codispoti B, Zamparini F, Zhang A, Esposti MD, et al. PLA-based mineral-doped scaffolds seeded with human periapical cyst-derived MSCs: A promising tool for regenerative healing in dentistry. Materials. 2019;12:1-17.

- line 140 - please describe in detail the manner in which exosomes were seeded on the scaffolds. Was the cell culture carried out only on the surface of the scaffold?

We did not seed the cell onto the scaffolds, we seeded only exosomes so we did not have to carried out the cells from the surfaces. The protocol of exosomes seeding has been inserted on MM section section 2.4

- line 306 - based on only in vitro cell growth studies, it is not possible to determine the biocompatibility of the material, this requires a much wider range of studies.

The biocompatibility of the biomaterials has been performed following the International standard guide line for biological test on biomaterials for dental application ISO 10993, so we think that the test we performed are sufficient to determine the in vitro biocompatibility of the material.  We changed the title of section 3.4 on :” 3.4. in vitro Biocompatibility of scaffolds”

We would like to clarify that several tests have been conducted on these scaffolds, including direct contact cytotoxicity (from Gandolfi 2018) and adhesion, growth, proliferation and osteogenic/odontogenic differentiation of human periapical cyst mesenchymal stem cells (Tatullo 2019).

Gandolfi MG, Zamparini F, Degli Esposti M, Chiellini F, Aparicio C, Fava F, et al. Polylactic acid-based porous scaffolds doped with calcium silicate and dicalcium phosphate dihydrate designed for biomedical application. Mater Sci Eng C. 2018;82:163-181.

Tatullo M, Spagnuolo G, Codispoti B, Zamparini F, Zhang A, Esposti MD, et al. PLA-based mineral-doped scaffolds seeded with human periapical cyst-derived MSCs: A promising tool for regenerative healing in dentistry. Materials. 2019;12:1-17.

- Figure 7 - in this type of research the results obtained with the participation of scaffold without minerals as well as on a neutral surface (e.g. TCPS) should be demonstrated as a reference test

Authors included tests on unloaded PLA and on plastic flasks.

It should be clarified that the aim of the article was no to compare the properties scaffolds with other scaffolds but to underline the properties of the exosomes able to enhance the properties of the scaffolds, so we don’t need any other reference test. 

- Figure 10 - here as a reference should be presented the results of tests on a neutral surface (e.g. TCPS)

Authors included tests on unloaded PLA and on plastic flasks.

It should be clarified that aim of the article was not to compare the properties scaffolds with other scaffolds but to underline the properties of the exosomes able to enhance the properties of the scaffolds, so we don’t need any other reference test.

- line 364 - the authors claim that the PLA matrix due to the rate of degradation is the most suitable in bone tissue engineering, because PLGA degrades too quickly. This is not true, as evidenced by the huge amount of work published so far, as well as the results of presented clinical and preclinical studies.

The discussion section has been modified ant the sentence has been removed. Information regarding the degradation rate of PLA, PGA and PCL were obtained from reference [Armentano et al. 2010], Table 1 reports the degradation rate of different synthetic polymers, showing that PLA degrades in 12-18 months, PCL after at least 24 months and PGA within 3-4 months.

- lines 376-380 - these are facts described and known, this information is not suitable for Conclusions part , but rather for the Introduction part.

The present paragraph has been moved in the introduction section

-The choice of material was unfortunate. In order for the work to make sense, it needs to be supplemented and rewritten, so as to emphasize the importance of exosomes, as well as the impact of mineral additives in the matrix. In the present form, the work is not suitable for publication, despite professionally conducted experiments.

The authors implemented introduction and discussion sections, as referees suggested.

-The authors clearly ignored the issues related to the proper selection of material and the method of forming optimal cell's scaffolds, which is extremely important if we want to obtain satisfactory results of conducted cultures and be able to compare them with the results of other researchers.

Reviewer 2 Report

The authors present an investigation on porous PLA-mineral-doped scaffolds associated with exosomes derived from a mesenchymal stem cell (MSC) on the osteogenic potential of MSC.

First, the authors provide the characterisation of the surface micromorphology two scaffolds produced with different amounts of CaSi and DCPD using ESEM and an  EDX analysis of the change in composition following 28 days immersion in HBSS.

Second, the presence of exosomes at the surface of the scaffold is observed using TEM and immunostaining.

Finally, cell cultures are performed to validate the interaction of the cells and the exosomes and to investigate the impact of scaffolds with or without exosomes on cell osteogenic commitment.

Although the investigations present a significant relevance with potentials application in regenerative medicine,  it suffers from major drawbacks.

Minor concerns

Several typos should be corrected for example line 135: 10.103cells/cm2, ligne 151: 10*106 cells: line 179: (Thermo-Fisher Scientific…..). line 292: figure 5 instead of 6

In addition, line 192: "192
whilst values comprised between -2 and 2 indicate differentially expressed genes" is inadequate

Major concerns

Methods:
2-the paragraph 2.4 should be moved after paragraph 2.6

3-through out the methods and results, it is not clear when the cells are seeded at the surface of the scaffold (as presented in fig7) or in the presence of exosomes released from the scaffold (as described in §2.4). The authors should be more rigorous in presenting the type of culture, indirect with exosomes released) or with direct scaffold/cell contact. The term (line 398)
placed in proximity of cultured hAD-MSCs is not clearly described in the methods (scaffold in an insert, or placed onto the cell at the confluence ?) and clearly impact the understanding of the cell culture conditions

Results

1- It is not clear why 28 days of immersion were chosen and why HBSS and not the culture medium as exosomes are realised from the scaffolds in culture mediua prior to addition to MSC

2- Figure 8: here, there are no approaches that allow to distinguish between cell surface adhesion of the exosomes and internalisation. Please provide further arguments on their internalisation

3-statistical analyses are missing in figures 7 and  10

4- line 317: "Exosomes released by labeled cells were also labeled upon fusion of multivesicular bodies with  the cell plasma membranE"

what type of label for the cells? cell labelling was not described in methods

5-figure 10: the appropriate reference will be the scaffold without exosomes and not cell on in plastic dished. where the cell-cultured on standard plastic dish also exposed to exosomes,  the authors should clearly indicate the culture conditions

Discussion :

1-it is not clear why exosomes extracted from hAD-MSC first, and them added to the culture of the identical cell should promote their osteogenic commitment. The authors should please elaborate on this aspect.

2-line 365: "other polymers (such as PLGA) degrades too rapidly (3-4 months) and others (such as PCL) degrades too slowly [17,18,24]" according to reference 17:  PLLA and PCL degradation time is >24 months. this sentence is then inadequately referring to it and should be corrected

3-Between the two different scaffolds  produced, the authors should elaborate on the one hey would choose  for further investigations

Author Response

REFEREE 2

The authors present an investigation on porous PLA-mineral-doped scaffolds associated with exosomes derived from a mesenchymal stem cell (MSC) on the osteogenic potential of MSC.

First, the authors provide the characterisation of the surface micromorphology two scaffolds produced with different amounts of CaSi and DCPD using ESEM and an  EDX analysis of the change in composition following 28 days immersion in HBSS.

Second, the presence of exosomes at the surface of the scaffold is observed using TEM and immunostaining.

Finally, cell cultures are performed to validate the interaction of the cells and the exosomes and to investigate the impact of scaffolds with or without exosomes on cell osteogenic commitment.

Although the investigations present a significant relevance with potentials application in regenerative medicine,  it suffers from major drawbacks.

Minor concerns

-Several typos should be corrected for example line 135: 10.103cells/cm2, ligne 151: 10*106 cells: line 179: (Thermo-Fisher Scientific…..). line 292: figure 5 instead of 6

We thank the referee. Typos have been corrected.

-In addition, line 192: "192

We thank the referee. Typos have been corrected.

-whilst values comprised between -2 and 2 indicate differentially expressed genes" is inadequate

The sentence has been modified “ as -2 and 2 indicate indifferentially expressed genes”

Major concerns

Methods: 
2-the paragraph 2.4 should be moved after paragraph 2.6

Paragraph has been moved

3-throughout the methods and results, it is not clear when the cells are seeded at the surface of the scaffold (as presented in fig7) or in the presence of exosomes released from the scaffold (as described in §2.4). The authors should be more rigorous in presenting the type of culture, indirect with exosomes released) or with direct scaffold/cell contact. The term (line 398) placed in proximity of cultured hAD-MSCs is not clearly described in the methods (scaffold in an insert, or placed onto the cell at the confluence ?) and clearly impact the understanding of the cell culture conditions.

Result section has been modified to be more clear.

Results

1- It is not clear why 28 days of immersion were chosen and why HBSS and not the culture medium as exosomes are realised from the scaffolds in culture mediua prior to addition to MSC

HBSS has been chosen as simulated body fluid, following a previously validated protocol which is used to assess materials apatite forming ability.

According with Gandolfi et al. 2010 “HBSS was selected to have a commercially available standardised soaking medium mimicking the composition of inorganic ions of human blood plasma, without representing a supersaturated solution with respect to apatite. HBSS contains lower calcium and same phosphate concentration than human plasma and was already used in bioactivity studies and in tissue engineering or culture cell culture studies “

Gandolfi, MG, Taddei P, Tinti A, Dorigo DE, Stefano E, Rossi PL, Prati C. Kinetics of apatite formation on a calcium silicate cement for root-end filling during ageing in physiological like phosphate solutions. Clin Oral Invest. 2010;14: 659-668.

Gandolfi MG, Zamparini F, Degli Esposti M, Chiellini F, Aparicio C, Fava F, et al. Polylactic acid-based porous scaffolds doped with calcium silicate and dicalcium phosphate dihydrate designed for biomedical application. Mater Sci Eng C. 2018;82:163-181.

2- Figure 8: here, there are no approaches that allow to distinguish between cell surface adhesion of the exosomes and internalisation. Please provide further arguments on their internalisation

Fig. 8 is not related to the internalization of the exosomes on the cell but is related to the presence of the exosomes onto the scaffolds surfaces. The internalization of the exosomes is evident on fig. 9

3-statistical analyses are missing in figures 7 and  10

Statistical analysis have been added in Fig 7 and 10

4- line 317: "Exosomes released by labeled cells were also labeled upon fusion of multivesicular bodies with  the cell plasma membranE" what type of label for the cells? cell labelling was not described in methods

We did not labeled cells, we labeled only exosomes isolated from non label cells and then EVs isolated were marked with PKH26 (Red Fluorescent Cell Linker Kits MINI26; Sigma-Aldrich Co., St Louis, MO, USA)

5-figure 10: the appropriate reference will be the scaffold without exosomes and not cell on in plastic dished. where the cell-cultured on standard plastic dish also exposed to exosomes,  the authors should clearly indicate the culture conditions

Stem cells were seeded in the plate, in presence of  growth medium MSCGM™

Discussion :

1-it is not clear why exosomes extracted from hAD-MSC first, and them added to the culture of the identical cell should promote their osteogenic commitment. The authors should please elaborate on this aspect.

The discussion has been improved following referee suggestions

2-line 365: "other polymers (such as PLGA) degrades too rapidly (3-4 months) and others (such as PCL) degrades too slowly [17,18,24]" according to reference 17:  PLLA and PCL degradation time is >24 months. this sentence is then inadequately referring to it and should be corrected

The discussion section has been modified ant the sentence has been removed. Information regarding the degradation rate of PLA, PGA and PCL were obtained from reference [Armentano et al 2010], Table 1 reports the degradation rate of different synthetic polymers, showing that PLA degrades in 12-18 months, PCL after at least 24 months and PGA within 3-4 months.

3-Between the two different scaffolds  produced, the authors should elaborate on the one they would choose  for further investigations

The author modified the text according with referee suggestions.

Round 2

Reviewer 1 Report

The authors redrafted the manuscript text in accordance with my comments. In most cases, they introduced signaled additions and explanations. The current version clearly shows that the main target of the presented work was to demonstrate the possibility of using exosomes to enhance in bone cell culture carried out on scaffolds. In this perspective, the properties of scaffold and the material from which it was made are not particularly important.

However, in the text is many errors and inaccuracies, which must be eliminated.

Title - "polilactide acide" is a rather incorrect term and confusing.  According to the terminology of IUPAC you should use term – “poly(L-lactide)”.

Line 39 - in biomedical applications, the term “eco-friendly” is at least missed - we should rather use the term “bioresorbable” or “biocompatible and biodegradable”. This should be taken into account throughout the manuscript text.

Line 84 - The description regarding polylactide should be significantly shortened. Please limit yourself to the most important data.

  In my opinion, the whole of Introduction part should be presented in a much more compressed form.

Line 143 - the fact that lactic acid (substrate to lactide manufacturing) is obtained by fermentation of polysaccharides (non by composting) is not important here, and better remove this information.  

Line 406 - please provide the characteristics of the "plastic flask" used in comparative studies (e.g. surface cell culture flask made with polystyrene sterile; g - irradiated)

Line 610 - This observation is superfluous and quite annoying. The usefulness of polylactides and, above all, L-lactide copolymers in scaffolding formation was proved much earlier.

At the same time, for the future, my friendly advice to authors - please use polymer materials - which are approved for biomedical applications. Only then can they be sure of the full reliability of the results obtained, and these results will not be undermined by others.

Author Response

The authors redrafted the manuscript text in accordance with my comments. In most cases, they introduced signaled additions and explanations. The current version clearly shows that the main target of the presented work was to demonstrate the possibility of using exosomes to enhance in bone cell culture carried out on scaffolds. In this perspective, the properties of scaffold and the material from which it was made are not particularly important.

However, in the text is many errors and inaccuracies, which must be eliminated.

Title - "polilactide acid" is a rather incorrect term and confusing. According to the terminology of IUPAC you should use term – “poly(L-lactide)”.

Thank you, the title has been corrected.

Line 39 - in biomedical applications, the term “eco-friendly” is at least missed - we should rather use the term “bioresorbable” or “biocompatible and biodegradable”. This should be taken into account throughout the manuscript text.

Thank you, “eco friendly” term has been changed as suggested.

Line 84 - The description regarding polylactide should be significantly shortened. Please limit yourself to the most important data.

Description has been shortened according with referee suggestion

In my opinion, the whole of Introduction part should be presented in a much more compressed form.

Introduction has been shortened, according with referee suggestion.

Line 143 - the fact that lactic acid (substrate to lactide manufacturing) is obtained by fermentation of polysaccharides (non by composting) is not important here, and better remove this information.

The information has been removed.

Line 406 - please provide the characteristics of the "plastic flask" used in comparative studies (e.g. surface cell culture flask made with polystyrene sterile; g - irradiated)  

“Plastic flasks” characteristics have been added in paragraph 2.7 of Matherials and methods

Line 610 - This observation is superfluous and quite annoying. The usefulness of polylactides and, above all, L-lactide copolymers in scaffolding formation was proved much earlier.

Sentence has been removed according with referee suggestion.

At the same time, for the future, my friendly advice to authors - please use polymer materials - which are approved for biomedical applications. Only then can they be sure of the full reliability of the results obtained, and these results will not be undermined by others.
